**Data Availability Statement:** All relevant data are within the manuscript and its Supporting Information files.

**Funding:** The authors received no specific funding for this work.

# Quality of life of breast cancer patients in Amhara region, Ethiopia: A cross-sectional study

**Tamrat Alem**[1]*, **Dabere Nigatu**[2], **Amsalu Birara**[3], **Tamene Fetene**[4], **Mastewal Giza**[5]

1 Felege Hiwot Regional Referral Hospital, Bahir Dar, Ethiopia, 2 Department of Reproductive Health and Population Studies, School of Public Health, College of Medicine and Health Sciences, Bahir Dar University, Bahir Dar, Ethiopia, 3 Department of Environment Health Sciences, School of Public Health, College of Medicine and Health Sciences, Bahir Dar University, Bahir Dar, Ethiopia, 4 Department of Pediatrics Nursing, College of Medicine and Health Sciences, Wolkite University, Wolkite, Ethiopia, 5 Department of Nutrition, College of Medicine and Health Sciences, Woldia University, Woldia, Ethiopia

* tamratalem34@gmail.com

## Abstract

### Background

Although breast cancer has a markedly higher incidence in developed countries, seven out of ten deaths occur in developing countries, including Ethiopia. However, there is a limited information on the quality of life (QoL) among breast cancer patients in Ethiopia, notably in the Amhara region. Therefore, this study aimed to assess the QoL and its associated factors among patients with breast cancer in the Amhara Region, Ethiopia.

### Methods

An institutional based cross-sectional study was conducted from 25th March 2019 to 7th July 2019. A systematically selected sample of 256 breast cancer patients were participated in the study. A standardized interviewer-administered Amharic version questionnaire was used to collect the data. We used the European Organization for Research and Treatment of Cancer quality of life questionnaire core 30 (EORTC QLQ C30) and breast cancer supplementary measure (QLQ-BR23) to measure QoL. The data were analyzed by SPSS version 23. A binary logistic regression model was fitted to identify the predictors of QoL. The adjusted odds ratio (AOR) with a 95% confidence interval (CI) was reported to show the strength of the association.

### Results

Sixty-eight percent of breast cancer patients had poor QoL (68.4%; 95% CI: 62.5–73.8). The mean score of QoL was 70.6 (standard deviation (SD) ±13.9; 95% CI: 69.0–72.4). All functional component scores were less than 75 on the symptom scale. Diarrhea (11.6), constipation (17.5), and dyspnea (24.7) were less noticeable symptoms. Being out of marriage (AOR = 2.59, 95% CI: 1.32–5.07), being poor (AOR = 2.39, 95%CI: 1.32–5.03), being non-housewife (AOR = 3.25, 95% CI: 1.16–7.22), and being complaints of dyspnea (AOR =

**Competing interests:** The authors have declared that they have no competing interests.

3.48, 95% CI: 1.79–6.79), and insomnia (AOR = 2.03, 95% CI: 1.05–3.91) were significantly associated with QoL.

## Conclusions

The proportion of poor QoL among breast cancer patients was high. Health care professionals should give attention to breast cancer patients who are out of marriage, poor and non-housewife while offering the recommended treatment courses.

## Introduction

Breast cancer is the leading cause of cancer-related mortality among women worldwide [1–5]. Globally, in 2018, it was estimated that 627,000 women died from breast cancer, which accounts for 15% of all cancer deaths among women. Approximately, 2.1 million new cases were diagnosed per year [5]. In Ethiopia, breast cancer is the most prevalent type of cancer and the leading cause of mortality among women [6] and accounts for 34% of all female cancer cases [7]. In 2018, over 15,000 breast cancer cases were diagnosed, and an estimated 8,000 cases died [5]. In Ethiopia, 15,244 (32.9%) new breast cancer cases were diagnosed, and the incidence and mortality rate of breast cancer were 42 and 23 per 100,000 respectively [8].

The World Health Organization (WHO) defines the quality of life (QoL) as "an individual's perception of their position in life in the context of the culture and value systems in which they live and about their goals, expectations, standards, and concerns." It is a broad-ranging concept affected in a complex way by the person's physical health, psychological state, personal beliefs, social relationships, and their relationship to salient features of their environment [9].

The consequences of breast cancer in the poorest settings are socially and economically devastating [10]. QoL plays an increasingly important role in treatment decisions, and it has become an obligatory aspect of evaluating new treatments [11]. Better QoL has been associated with longer survival in patients with cancer [12].

The impairment in the QoL starts with the diagnosis of cancer and continues with the aggressive nature of treatment [13]. QoL in breast cancer is influenced by the disease itself (direct disease effects, stage at diagnosis, and clinical course), the treatment of the disease [14], comorbidity [15], age at presentation, race or ethnicity, and socioeconomic status. Oncologic medical treatments may lead to QoL improvements, but sometimes a wide variety of side effects can arise, bringing about significant health-related complaints [16]. The most common systematic chemotherapeutic side effects are nausea, and vomiting, followed by fatigue [14, 17], which can be emotionally distressing and debilitating, which in turn may affect their QoL. Women receiving chemotherapy and took more than three cycles of chemotherapy had lower QoL [18, 19].

Studies revealed that ages 51 to 60 years [20], a young age [21], and unmarried patients [20, 22] were associated with poor QoL. On the other hand, higher household income [23], older than 55 years of age, post-menopausal, stage I malignancy, patients who have completed treatment, and patients who underwent breast-conserving surgery were more likely to have a better QoL [24]. A study done in Tikur Anbessa Hospital also revealed that patients with advanced education had good QoL [19]. Today, QoL can significantly impact the diagnosis and treatment course of breast cancer, resulting in a better prognosis among patients [25], and QoL is an important outcome criterion in oncology [26].

Though breast cancer is the leading cause of morbidity and mortality among women in Ethiopia, we accessed only a few published studies that addressed the QoL of breast cancer patients [19]. The majority of the accessed studies also addressed only limited treatment courses or categories. Therefore, this study assessed the QoL and its associated factors among breast cancer patients in the Amhara region, Ethiopia. The study included breast cancer patients in various treatment categories, such as surgery, chemotherapy, and post-treatment follow-up to get a broad picture of the QoL. Measuring the proportion of poor QoL and identifying the factors associated with it may provide insight for health system planners, intervention programmers, and healthcare professionals into how to improve the QoL of breast cancer patients and, their survival in Ethiopia.

## Materials and methods

### Study setting and population

An institution-based cross-sectional study design was employed in the oncology centers of three public hospitals in the Amhara regional state, Ethiopia. The three public hospitals considered in the study were Felege Hiwot Comprehensive Specialized Hospital (FHCSH), Gondar Referral Teaching Hospital (GRTH), and Dessie Referral Hospital (DRH). FHCSH is located in Bahir Dar, 565 kilometers away from Addis Ababa, the capital city of Ethiopia. The hospital started oncology services in April 2016 with 18 inpatient beds. GRTH is located in Gondar town 750 kilometers far from Addis Ababa, and it started to provide oncology services in January 2015 with 17 inpatient beds. DRH is located in Dessie town and it started providing oncology services as of October 2017 with 15 inpatient beds. Currently, three of the hospitals have both outpatient and inpatient departments for cancer diagnosis and treatment, including surgery and chemotherapy services; however, none of the hospitals have radiotherapy service. In 2018/19, there were 217, 191, and 168 breast cancer patients on treatment or post-treatment follow-up in FHCSH, GRTH, and DRH, respectively.

The source populations were those breast cancer patients who were evaluated and treated in the oncology units of the three hospitals. Those breast cancer patients who visited the hospitals and were evaluated or treated at the oncology units from 25th March 2019 to 7th July 2019 were the study population. We included female breast cancer patients aged 18 years or older and who had received at least two or more cycles of chemotherapy, who were on post-treatment follow up, or who received surgical therapy irrespective of receiving chemotherapy while excluding those patients who had known cases of chronic illness or those patients who were newly diagnosed for breast cancer.

### Sample size and sampling technique

The sample size was determined using a single population proportion formula: $n = \frac{(Z_{\alpha/2})^2 P(1-p)}{d^2}$. The following assumptions were taken into consideration: a 95% confidence level ($Z_{\alpha/2}$ = 1.96), 5% marginal error (d = 0.05), and an 80% proportion of poor QoL of breast cancer patients [27]. Then, adding 10% to compensate for non-response (246/(1−0.1) = 273), the calculated sample size was 273.

A systematic random sampling technique was used to select study participants. The sample was taken proportionally to each hospital patient load. The previous three months patient load were taken from patient logbook of each hospital (i.e., FHCSH = 217, GRTH = 191, and DRH = 168; total load = 576). A sampling interval (K) of 2 was applied to select patients. A lottery method was used to select the first patient and then, every other patient was included according to their order of visit to the oncology unit.

## Data collection and variable measurement

Data were collected through face-to-face interview and patient's chart review. Six trained BSc Nurses were collected the data under the supportive supervision of three BSc Nurses. A structured questionnaire and data extraction checklist were used to collect socio-demographic, economic, clinical, and QoL data. Medical data such as the stage of the disease, type of treatment, type of surgery, cycles of chemotherapy, and other medical conditions were extracted from the patient's medical charts. The socio-demographic data included residence, age, education, religion, occupation, and marital status. The economic status was measured by the wealth index. The wealth index was assessed separately for rural and urban residents. The tool used to assess the wealth index was adapted from the 2016 Ethiopian demographic and health survey questionnaire.

The quality of life of breast cancer patients was the outcome variable for this study. We used the European Organization for Research and Treatment of Cancer Quality of Life Questionnaire Core 30 (EORTC QLQ C30) and breast cancer supplementary measure (QLQ-BR23) to measure QoL. The Amharic version of EORTC QLQ version 3 of QLQ-C30 and its breast cancer supplementary measure (QLQ-BR23) were accessed and used after getting permission from the EORTC. This tool is a disease-specific QoL scale. In the assessment of patients' QoL, disease-specific QoL scales are preferred because they are sensitive and are capable of detecting small but clinically significant changes in health [28]. The QoL measurement scales used were a reliable and valid measure of QoL in cancer patients; the internal consistency had a Cronbach's α value of 0.81. In the internal consistency of the subscale, a Cronbach's α value was greater or equal to 0.73 except for cognitive function (Cronbach's α = 0.29) [29]. Another study in Ethiopia, Tikur Anbessa Specialized and Referral Hospital, shows satisfactory internal consistency (Cronbach's α coefficients > 0.7), except for cognitive function (α = 0.52) of EORTC QLQ-C30 and body image (α = 0.51) of EORTC QLQ-BR23. Multiple-trait scaling analysis demonstrated a good convergent and divergent validity. Most items in EORTC QLQ-BR23 had a weak or no correlation with their own dimension in EORTC QLQ-C30 (r < 0.4) except for some symptom scales. A statistically significant change in quality of life scores (P ≤ 0.05) was observed in all dimensions of both instruments between baseline and the end of first cycle chemotherapy, except for body image (P = 0.985) and sexual enjoyment (P = 0.817) scales of EORTC QLQ-BR23, indicating clinical validity[30].

The EORTC QLQ-C30 is a tool used to address quality of life issues to all cancer type patients and it is composed of nine multi-item scales and six single items. The multi-item scales include five functioning scales (physical, role, cognitive, emotional, and social), a global health status (QoL) scale, and three symptom scales (fatigue, pain, and nausea/vomiting). The six single items included dyspnea, insomnia, appetite loss, constipation, diarrhea, and financial difficulties [31]. The EORTC QLQ-BR23 is unique to breast cancer patients and it is composed of four functional scales (future perspective, body image, sexual function, and sexual enjoyment) and four symptom scales (systemic therapy side effect, arm symptom, breast symptom, financial difficulties, and upset by hair loss). The global health status (QoL) had two questions, with a modified 7-point linear analog scale ranging from 1 "very poor" to 7 "excellent". All other items are scored on a 4-point categorical scale ranging from 1 "not at all" to 4 "very much".

There is no agreed threshold score to mean significant impairment for the EORTC QLQ-C30 and QLQ-BR23 scales. However; a study in Ethiopia, Tikur Anbessa Specialized and Referral Hospital, dichotomized each scale and subscales into the "good" or "poor" category [32]. We have followed the same classification for the current study.

### In functional component/scale and global health status or QoL

✓ **Good**- Higher scores on the functioning/global health status scale75 and above

✓ **Poor**- lower mean score in the functioning/global health status scales (75 and lower)

### In symptom scale/item

✓ **Good**- when the mean score is lower or less than 25

✓ **Poor**- when the mean score is higher or 25 and above

### Data analysis

The data were coded and entered into EPI data version 3.1 software. Then, exported to statistical packages for social sciences (SPSS) version 23 software for further analysis. Descriptive statistics were used to summarize the data in the form of frequency, mean, standard deviation (SD), and cross-tabulation. The internal consistency of the EORTC QLQ was evaluated using the reliability coefficient (i.e., Cronbach's alpha value). The Cronbach's alpha value of EORTC QLQ-C30 and QLQ-BR23 was 0.80. The reliability coefficient of each subscale was also greater than 0.7 except for the cognitive function (0.63) and pain (0.65) subscales. We used Hosmer-Lemeshow's goodness-of-fit test to evaluate model fitness, its p-value was 0.68.

The EORTC QLQ-C30 and QLQ-B23 scoring manual were used to create raw scores and transform the raw scores to 0 to 100 values [33]. A 100 corresponds to the maximum score while 0 corresponds to the minimum score. A high scale score represents a higher response level. Thus, a high score for a functional scale or a global health status/QoL scale represents a high/healthy level of functioning while a high score for a symptom scale/item represents a high level of symptomatology/problems. The principle for scoring these scales is the same in all cases and it involves two procedures: 1) Raw Score, which is the average of the items that contribute to the scale, and 2) A linear transformation to standardize the raw score [33].

The procedure for these calculations presented as follows:

$$\text{The raw score(RS)} = (I_1 + I_2 + I_3 + \cdots + I_n)/n$$

Where, $I_1 + I_2 + I_3 + \ldots + I_n$, are items included in the scale

A linear transformation: we applied the linear transformation to 0–100 to obtain the score $S$,

Functional scale : $S = \left\{ 1 - \frac{RS-1}{Range} \right\} x100$

Symptom scale/item : $S = \left\{ \frac{RS-1}{Range} \right\} x100$

Global health status/QoL : $S = \left\{ \frac{RS-1}{Range} \right\} x100$

Where; a range is a difference between the maximum possible value of RS and the minimum possible value. All items of any scale in the QLQ-C30 and QLQ-BR23 have been designed to take the same range of values. Therefore, the range of RS equals the range of item values. Most items are scored 1 to 4, giving a range = 3. The exceptions are the items contributing to the global health status/QoL, which are 7-point scale questions with a range = 6.

Bi-variable and multivariable binary logistic regression analyses were carried out to identify factors associated with the outcome variable. Variables with P-value less than 0.2 in bi-variable logistic regression were considered to fit the multivariable logistic regression model. A p-value of less than 0.05 was used to determine the presence of a significant association in the

multivariable logistic regression model. Wealth index for rural and urban residencies was separately analyzed by principal component analysis.

## Ethical consideration

Ethical approval (CMHS/IRB 03–008) letter was obtained from the Institutional Review Board (IRB) of the College of Medicine and Health Sciences, Bahir Dar University. The administrators of each hospital were communicated with an official letter and we got permission from each official to go ahead with the study. We obtained written informed consent from each participant according to the Helsinki declaration. Patients' privacy and confidentiality of information were maintained throughout the study process.

## Results

### Socio-demographic characteristics of patients

A total of 256 breast cancer patients were interviewed, which gives a response rate of 93.8%. The participants' mean (SD) age was 44.34 (± 14.11) years, with a range of 22 to 95 years. One hundred fifty-five (60.5%) patients were married. One hundred fifty-three (59.8%) were urban residents, and 190 (74.2%) were Orthodox Christians. About seventy percent of the patients were house-wives, and 134 (52.3%) had no formal education. One hundred sixteen (45.3%) patients had health insurance schemes to cover the cost of treatment, and 91 (35.6%) patients were from poor wealth (Table 1).

### Clinical characteristics of the patients

One hundred eighty-two (70.6%) (Stage III, 38.3%, and stage IV, 32.3%) patients were with advanced stages of breast cancer, and 96.9% of the patients receiving/received chemotherapy

**Table 1. Socio-demographic characteristics of breast cancer patients, Amhara region, Ethiopia, 2019 (N = 256).**

| Variables | Categories | Frequency | Percentage |
|---|---|---|---|
| **Age** | ≤40 years | 129 | 50.4 |
| | 41–54 years | 64 | 25.0 |
| | ≥55 years | 63 | 24.6 |
| **Residence** | Urban | 153 | 59.8 |
| | Rural | 103 | 40.2 |
| **Marital status** | Married | 155 | 60.5 |
| | Unmarried | 101 | 39.5 |
| **Educational status** | No formal education | 134 | 52.3 |
| | Primary (1–8) | 46 | 18.0 |
| | Secondary (9–12) | 24 | 9.4 |
| | Higher | 52 | 20.4 |
| **Religion** | Orthodox | 190 | 74.2 |
| | Muslim/protestant | 66 | 25.8 |
| **Occupation** | House-wife | 180 | 70.3 |
| | Non-housewife [a] | 76 | 29.7 |
| **Cost of treatment** | Health insurance/Free | 140 | 54.7 |
| | Private/self | 116 | 45.3 |
| **Wealth index** | Poor | 91 | 35.6 |
| | Medium | 83 | 32.4 |
| | Rich | 82 | 32.0 |

[a]student, farmer and daily laborer, merchant, gov't employee

treatments. Seventy-eight percent of patients were less than 12 months since they were diagnosed with breast cancer. The mean (±SD) duration from diagnosis was 12± (12.6) months (Table 2).

## Quality of life of breast cancer patients

The mean score for the global health status (QoL) for breast cancer patients was 70.6 (SD = 13.9, 95% CI: 69.0–72). The mean scores on the functional scale range from 43.8 (SD = 35.2) for emotional function to 64.2 (SD = 27.7) for social function. In the symptom scale, almost all symptoms were noticeable with different levels of intensity. The mean score of symptom scales ranged from as high as 67.2 (SD = 34.3) for appetite loss to as low as 11.6 (SD = 25.6) for diarrhea (Table 3).

On the EORTC QLQ-BR23, mean scores on functional scales ranged from 40.5 (SD = 42.5) for future perspective to 67.5 (SD = 33.3) for sexual enjoyment. Mean scores on symptoms scales ranged from 57.0 (SD = 41.6) for an upset by hair loss to 63.0 (SD = 34.1) for breast symptoms (Table 3).

One hundred seventy-five (68.4%) of breast cancer patients' QoL was poor. From the functional status perspective, most patients had a poor emotional function, physical function, and future perspective 207 (80.9%), 183 (71.5%), and 183 (71.5%), respectively. One hundred twenty-two (47.7%) sexual function and 109 (42.6%) body image of the patient's QoL was good (Fig 1).

As shown in the graph below from the symptom scale; most of the breast cancer patients were affected by appetite loss, 226 (88.3%), and systemic therapy side effects, 219 (85.5%). However, breast cancer patients were less affected by diarrhea (54 (21.1%)) and constipation (94 (36.7%)) (Fig 2).

## Factors affecting the quality of life of breast cancer patients

The results of the multivariable logistic regression analyses showed that marital status, wealth status, insomnia, and dyspnea were significant factors that affect the QoL of breast cancer patients.

**Table 2. Clinical characteristics of the breast cancer patients, Amhara region, Ethiopia, 2019 (N = 256).**

| Variables | Categories | Frequency | Percent (%) |
|---|---|---|---|
| Stage of breast cancer | Early stage | 74 | 28.9 |
| | Advanced stage | 182 | 71.1 |
| Duration of disease | ≤12 months | 200 | 78.1 |
| | 13–24 months | 28 | 10.9 |
| | 25–36 months | 13 | 5.1 |
| | > 36 months | 15 | 5.9 |
| Type of treatment | Chemotherapy +surgery | 217 | 84.8 |
| | Chemotherapy only | 31 | 12.1 |
| | Surgery only | 8 | 3.1 |
| Cycle of Chemotherapy | 1-3cycle | 108 | 42.2 |
| | 4–6 cycle | 83 | 32.4 |
| | ≥7cycle | 51 | 19.9 |
| | Completed | 14 | 5.5 |
| Type of surgery | Mastectomy | 205 | 91.1 |
| | Conserving | 20 | 8.9 |

**Table 3. Mean and standard deviation of EORTC QLQ-C30 and BR23, components for breast cancer patients, Amhara region, Ethiopia, 2019 (N = 256).**

| | | Analysis Category | Question N° | Mean | SD | 95% CI |
|---|---|---|---|---|---|---|
| EORTC QLQ-C30 | Global health/QoL | | 29 & 30 | 70.6 | 13.9 | 69.0–72.3 |
| | Functional scale | Physical function | 1–5 | 52.4 | 33.8 | 48.0–56.2 |
| | | Role function | 6 & 7 | 61.8 | 25.6 | 58.7–65.0 |
| | | Emotional function | 21–24 | 43.8 | 35.2 | 39.6–48.2 |
| | | Cognitive function | 20 & 25 | 62.2 | 31.4 | 58.5–66.2 |
| | | Social function | 26 & 27 | 64.2 | 27.7 | 60.9–67.7 |
| | Symptom scale | Fatigue | 10, 12 & 18 | 60.5 | 31.9 | 56.3–64.5 |
| | | Nausea & vomiting | 14 & 15 | 40.9 | 37.0 | 36.3–45.8 |
| | | Pain | 9 & 19 | 53.6 | 30.3 | 49.7–57.5 |
| | | Dyspnea | 8 | 24.7 | 29.9 | 21.2–28.3 |
| | | Insomnia | 11 | 34.1 | 35.3 | 29.8–38.5 |
| | | Appetite loss | 13 | 67.2 | 34.3 | 63.2–71.2 |
| | | Constipation | 16 | 17.5 | 26.2 | 14.3–20.8 |
| | | Diarrhea | 17 | 11.6 | 25.6 | 8.6–15.1 |
| | | Financial difficulties | 28 | 63.3 | 39.4 | 58.5–68.1 |
| EORTC-QLQ-BR23 | Functional scale | Body image | 39–42 | 64.3 | 33.3 | 60.1–68.1 |
| | | Sexual function | 44 & 45 | 67.5 | 33.3 | 63.6–71.7 |
| | | Sexual enjoyment (N = 132) | 46 | 46.5 | 29.6 | 41.4–51.8 |
| | | Future perspective | 43 | 40.5 | 42.5 | 34.9–46.0 |
| | Symptom scale/items | Systemic therapy side effects | 31–34 & 36–38 | 58.2 | 28.8 | 54.5–61.5 |
| | | Breast symptoms | 50–53 | 63.0 | 34.1 | 58.6–67.2 |
| | | Arm symptoms | 47–49 | 59.5 | 34.2 | 55.2–63.8 |
| | | Upset by hair loss (N = 231) | 35 | 57.0 | 41.6 | 51.4–62.5 |

Unmarried breast cancer patients were 2.59 times more likely to have poor QoL compared to married breast cancer patients (AOR = 2.59, 95% CI: 1.32–5.07). Breast cancer patients who were poor wealth status were 2.39 times more likely to have poor QoL compared to patients who were rich (AOR = 2.39, 95% CI: 1.32–5.03). Those non-housewives breast cancer patients were 3.25 times more likely to have poor QoL as compared to housewives (AOR = 3.25, 95% CI: 1.46–7.22). Breast cancer patients who had complaints of dyspnea were 3.48 times more likely to have poor QoL (AOR = 3.48, 95% CI: 1.79–6.79) compared to patients who had no complaints of dyspnea. Similarly, patients with complaints of insomnia were 2.03 times more likely to have poor QoL (AOR = 2.03, 95% CI: 1.05–3.91) compared to patients with compliant of insomnia (Table 4).

## Discussion

This study showed that 68.4% (95% CI: 62.5–73.8) breast cancer patients' QoL was poor. The mean score of QoL was 70.6 (95% CI: 69.0–72.4). The study identified that marital status, occupation, wealth status, insomnia, and dyspnea as predictors of poor QoL in breast cancer patients in the Amhara region, Ethiopia.

The proportion of poor QoL was 68.4%. This finding is similar with a study done in Addis Ababa (68.1%) [34]. However, this finding is lower than studies done in Addis Ababa (92.52%) [35], and India (79.8%) [27]. The difference might be due to a difference in study population with the Indian study, which included male breast cancer patients.

This is higher than the EORTC QLQ-C30 reference value 61.8(±24.6) [33], and other studies done in India (59.3), Malaysia (65.7±21.4), Morocco (68.5±18.5), Cote d'var and Addis

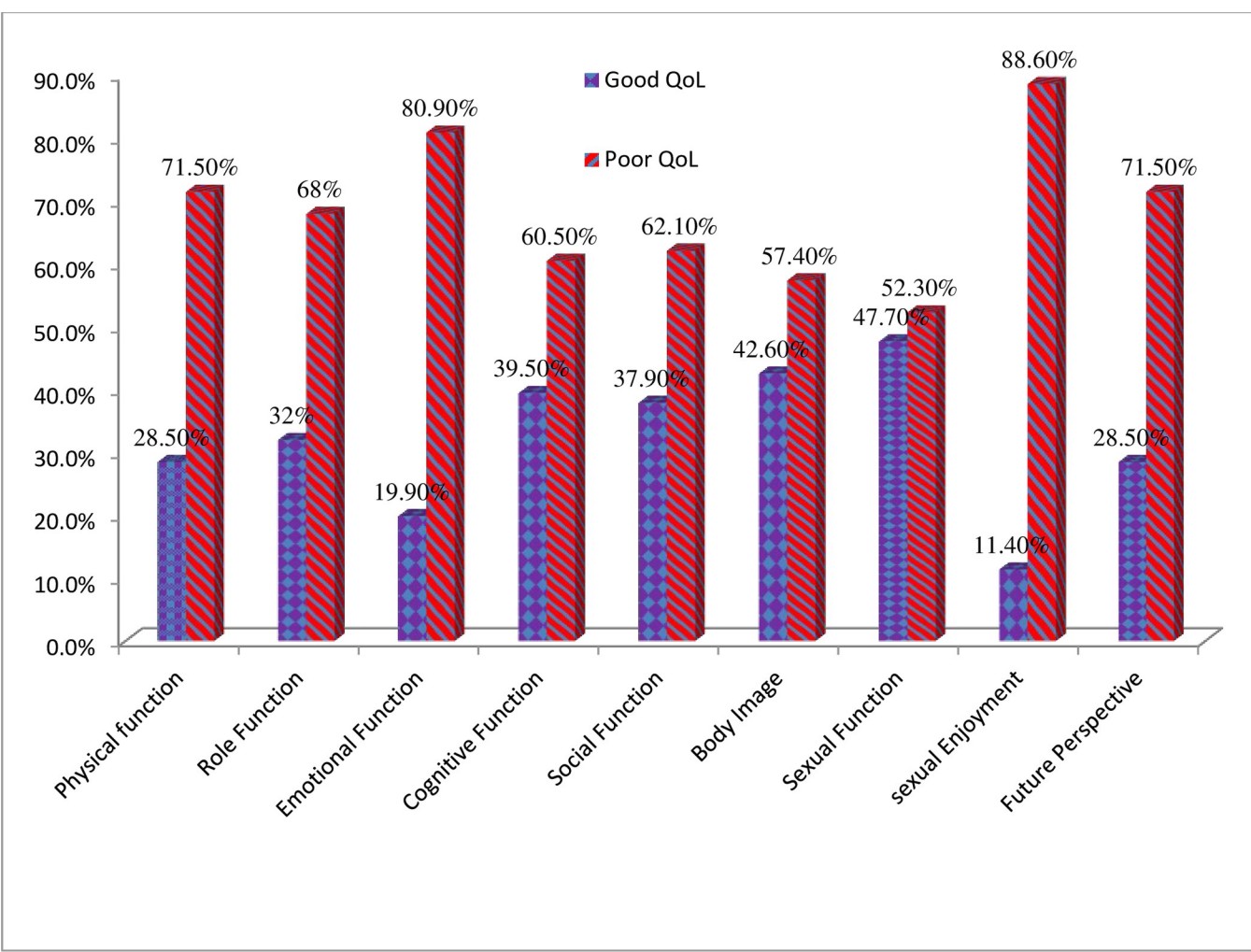

**Fig 1. Proportion of QoL by functional components among patients with breast cancer, Amhara region, Ethiopia, 2019.**

Ababa (53±25.6) [19, 20, 22, 24, 36]. In Malaysian, chronic illness patients were included, in Morocco, patients who received treatment for more than 3 months and patients with severe neuropsychiatric disorders patients were excluded. Paleri et al noted that breast cancer patients with comorbidities had reduced QoL [15]. The Addis Ababa study also included only breast cancer patients under chemotherapy course of treatment [19] while our study included patients across various courses of treatments. However, the mean score of QoL was lower than Colombia 77.5(±20.1). The possible reason might be due to socio-demographic differences such as a mean age of 55.7, 95% of the women reporting religious affiliation and relatively high level of socioeconomic status [37], all of which may be subject to improve the QoL. Although religious affiliation was not included in this study, being religious and high socioeconomic status improves QoL.

This study revealed that all functional components -physical function (52.3±33.8), role function (61.8±25.6), emotional (43.8±35.2), cognitive function (62.2±31.4) and social function (64.2±27.7) were lower than the EORTC QLQ C30 reference value 78.4±21.3, 70.9±29.9, 68.6±23.8, 81.5±21.8 and 77±27.1 respectively. Our findings are similar to a study done in India but lower than the studies done in Morocco, Malaysia, and Colombia [20, 22, 24, 37]. As

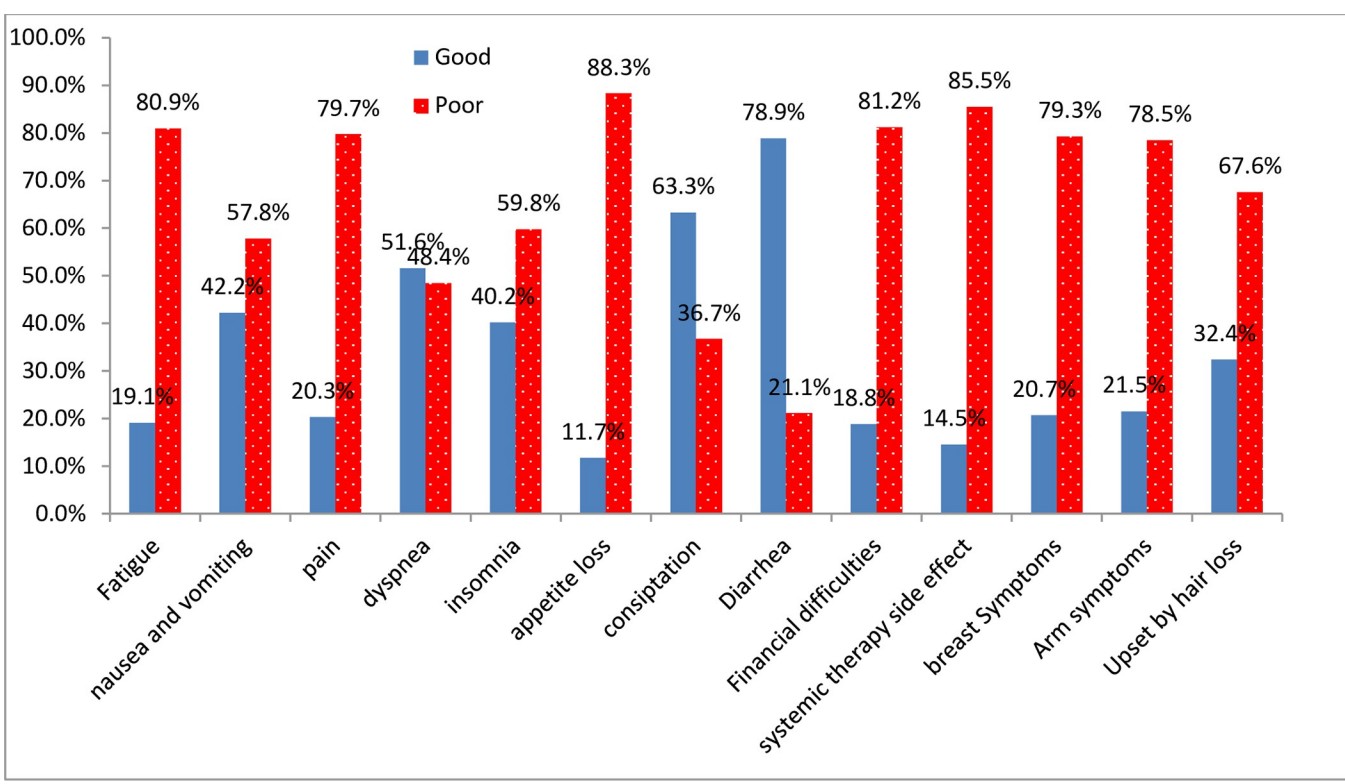

**Fig 2. Proportion of QoL by symptom scales/items among breast cancer patients, Amhara region, Ethiopia, 2019.**

compared to our study, the Malaysian study involved a smaller number of patients who were at the advanced stage of the disease (48% vs. 72%) and a small number of the patients who were receiving chemotherapy as a treatment option (38% vs. 96%). In Morocco, a few cases had stage IV (12.9%) breast cancer. This might be a possible reason because the side effects of chemotherapy and stages of the disease significantly affect the QoL in breast cancer patients.

From the EORTC QLQ C30, symptomatic scale/item constipation (17.5±26.3) was consistent with the EORTC QLQ-C30 reference value 17.4±27.2 [33] while fatigue (60.5±31.9), nausea and vomiting (40.9±37), pain (53.6±30.3), dyspnea (24.7±29.9), insomnia (34.1±35.3), appetite loss (67.2±34.3), diarrhea (11.6±25.6), and financial difficulties (63.3±39.4) were higher than the EORTC QLQ-C30 reference value. Diarrhea (11.6±25.6) consistent with a study done in Turkey [38]. In this study, 96.9% of breast cancer patients were receiving chemotherapy as a type of treatment. Ganz et al reported that patients receiving chemotherapy might experience several side effects that negatively affected their QoL [14].

From EORTC QLQ-BR23, body image and future perspective were lower than EORTC QLQ BR23 reference value [33] and a study done in Malaysia [24]. However, sexuality, systemic therapy side effects, and upset by hair loss, breast, and arm symptoms were higher than the EORTC QLQ BR23 reference value [33] and the Malaysian study [24]. This study revealed that women who underwent breast-conserving surgery had a better global health status than women who had a mastectomy. This finding is in line with other studies done in Morocco and Taiwan [22, 39]. This might be due to the fact that breast cancer patients who underwent mastectomy may start to worry about their body image and feel less attractive because of the surgery.

The current study found that being unmarried negatively affected the QoL of breast cancer patients. This is similar to studies done in Morocco, India, and Ethiopia [19, 20, 22]. Married patients tend to present early before metastasis and receive advanced care unlike unmarried

**Table 4. Factors associated with affected quality of life among breast cancer patients, Amhara region, Ethiopia, 2019 (N = 256).**

| Variables | Response | Quality of life (QoL) | | COR (95% CI) | AOR (95%CI) |
|---|---|---|---|---|---|
| | | Good | Poor | | |
| Marital status | Married | 62(40.0%) | 93(60.0%) | 1 | 1 |
| | Unmarried | 19(18.8%) | 82(81.2%) | 2.88(1.59–5.21) | 2.59(1.32–5.07)* |
| Wealth status | Poor | 22(24.2%) | 69(75.8%) | 2.58(1.35–4.93) | 2.39(1.32–5.03)* |
| | Medium | 22(26.5%) | 61(73.5%) | 2.28(1.19–4.38) | 1.90(0.88–4.08) |
| | Rich | 37(45.1%) | 45(54.9%) | 1 | 1 |
| Stages of disease | Early stage | 19(25.7%) | 55(74.3%) | 1.50(0.82–2.74) | 1.34(0.65–2.76) |
| | Advanced stage | 62(34.1%) | 120(65.9%) | 1 | 1 |
| Dyspnea | Poor | 23(18.5%) | 101(81.5%) | 3.44(1.95–6.08) | 3.48(1.79–6.79)*** |
| | Good | 58(43.9%) | 74(56.1%) | 1 | 1 |
| Insomnia | Poor | 38(24.8%) | 115(75.2%) | 2.17(1.27–3.71) | 2.03(1.05–3.91)* |
| | Good | 43(41.7%) | 60(58.3%) | 1 | 1 |
| Role function | Poor | 46(26.4%) | 128(73.6%) | 2.07(1.19–3.60) | 1.54(0.76–3.12) |
| | Good | 35(42.7%) | 47(57.3%) | 1 | 1 |
| Future perspective | Poor | 50(27.3%) | 133(72.7%) | 1.96(1.11–3.46) | 1.74(0.91–3.34) |
| | Good | 31(42.5%) | 42(57.5%) | 1 | 1 |
| Religious | Orthodox | 68(35.8%) | 122(64.2%) | 1 | 1 |
| | Muslim or protestant | 13(19.7%) | 53(80.3%) | 2.57(1.29–5.13) | 2.00(0.92–4.35) |
| Occupation | Housewives | 67(37.2%) | 113(62.8%) | 1 | 1 |
| | Not Housewives[a] | 14(18.4%) | 62(81.6%) | 2.63(1.37–5.05) | 3.25(1.46–7.22)** |

AOR: Adjusted odd ratio; CI: Confidence Interval; COR: Crude odd ratio

* P< 0.05

**P<0.01

*** p<001

[a]students, farmer, merchant, daily laborer

patients [32]. This might be because unmarried women who have been diagnosed with breast cancer might feel insecure about getting a partner and develop a fear of not being loved by others that likely compromise their QoL. There are also studies with null [19] or reversal associations with the current findings [24]. The presence of inconsistent findings appeals for further investigation.

The study also revealed that not being a housewife by occupation negatively influenced the QoL of breast cancer patients. This is similar with a study done in India [20]. The reason might be that disease and treatment-related side effects disrupted their daily lives, work schedules, and financial stability. The treatment requires frequent hospital visits and costs for transportation, diagnostics, treatment, and accommodation may worsen the QoL.

In this study, breast cancer patients with poor wealth status were found to have poor QoL. This is similar to a study done in Shanghai, China [18], and Asia [23]. This might be because of poor wealth status patients might be unlikely to access comprehensive care because of financial problems to cover direct (i.e., health care costs) and indirect costs such as transportation and accommodation costs. In most instances, chemotherapy medications, including strong analgesics for managing disease and treatment side effects and diagnostics, may not be available at government hospitals in the current study setup. These all incur additional costs for patients and likely affect their QoL. A study also indicated that financial problems are the most devastating for cancer patients; nearly 2 out of 3 patients may sell their homes/other household assets to cover medical care and other costs [40].

The most common symptoms on the symptom scale were dyspnea and insomnia. Both symptoms were associated with poor QoL. The mean score of insomnia was greater than 25, which is the most symptomatic and significantly affects the QoL. This is similar to a study done in Addis Ababa, Ethiopia [19]. In this study, pain was the most common complaint and there might not be adequate pain management and prescribed opioid analgesics. In this study, about 71% of breast cancer patients were at an advanced stage, which likely reduced their QoL. When breast cancer is at an advanced stage, it might metastasize to the lung and other organs which leads the patient to face difficulty breathing. The disease itself and treatment side effects can also result in patients having stress and disturbed sleeping patterns.

This study had both strengths and limitations. As a strength, the study considered main hospitals with oncology centers in the Amhara region, breast cancer patients at various treatment cycles and types, and various stages of the disease. This gave us the chance to observe a broad picture of the QoL issues in the Amhara region, Ethiopia.

As a limitation, some of the questions in the interviews were personal or sensitive issues; therefore, response bias is a possible limitation of the study. Because of the cross-sectional natures of the study design, data on QoL before the diagnosis or before starting the treatment were not available, and it was therefore not possible to assess the temporal relationship. Sample size calculation was considered with other similar country. Participants were also required to recall events as far back as a month before the interview, and therefore, recall bias is also a possible limitation.

## Conclusions

We conclude that the proportion of poor QoL among breast cancer patients in Amhara region was high. The study also identified that being unmarried, not being a housewife, having poor wealth status, and having complaints of insomnia and dyspnea significantly affected the QoL of breast cancer patients in the Amhara region, Ethiopia. The Ministry of Health and the Amhara Regional Health Bureau should work to incorporate QoL assessment and intervention component in the breast cancer treatment protocol. Development partners and charity organization should also give attention for breast cancer patients who live in financial hardship, and out of marriage to provide financial, psychological and emotional support. The Health Care Professionals should also recognize and take into consideration the importance of addressing the QoL issues of the patients aside the clinical treatment. Further studies with strong design, for example, prospective cohort studies, are recommended to identify the determinants of QoL.

## Supporting information

**S1 File. Amharic version questionnaire.**
(DOCX)

**S2 File. English version questionnaire.**
(DOCX)

**S1 Dataset. Study dataset.**
(SAV)

## Acknowledgments

The authors would like to acknowledge the IRB of the College of Medicine and Health Sciences at Bahir Dar University for their ethical review. The authors would like to thank the EORTC

for permitting to use the Amharic of the EORTC QLQ-C30 and BR23 scales. Furthermore, the authors would like to acknowledge FHCSH, DRH, and GRTH officials and the oncology centers staff members for their unlimited support and providing the required information. Finally, the authors thank data collectors, supervisors, and study participants for their unreserved support and willingness to participate during the data gathering process.

## Author Contributions

**Conceptualization:** Tamrat Alem, Amsalu Birara.

**Data curation:** Tamrat Alem, Dabere Nigatu, Amsalu Birara, Tamene Fetene, Mastewal Giza.

**Formal analysis:** Tamrat Alem, Dabere Nigatu, Amsalu Birara, Tamene Fetene, Mastewal Giza.

**Funding acquisition:** Tamrat Alem.

**Investigation:** Tamrat Alem.

**Methodology:** Tamrat Alem, Dabere Nigatu, Amsalu Birara, Tamene Fetene, Mastewal Giza.

**Project administration:** Tamrat Alem.

**Resources:** Tamrat Alem.

**Software:** Tamrat Alem, Dabere Nigatu, Amsalu Birara, Mastewal Giza.

**Supervision:** Tamrat Alem.

**Validation:** Tamrat Alem, Dabere Nigatu, Amsalu Birara, Tamene Fetene, Mastewal Giza.

**Visualization:** Tamrat Alem, Dabere Nigatu.

**Writing – original draft:** Tamrat Alem, Tamene Fetene.

**Writing – review & editing:** Tamrat Alem, Dabere Nigatu, Amsalu Birara, Tamene Fetene, Mastewal Giza.

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
