## [Decision Letter · Decision Letter 0]

2 Nov 2022

PONE-D-21-10111Quality of life and associated factors among patients with breast cancer in Amhara region, EthiopiaPLOS ONE

Dear Dr. Alem,

Thank you for submitting your manuscript to PLOS ONE. After careful consideration, we feel that it has merit but does not fully meet PLOS ONE’s publication criteria as it currently stands. Therefore, we invite you to submit a revised version of the manuscript that addresses the points raised during the review process.

We look forward to receiving your revised manuscript.

Kind regards,

Esmat Mehrabi

Academic Editor

PLOS ONE

Journal Requirements:

“Not applicable”

6. Please include a separate caption for each figure in your manuscript.

Reviewers' comments:

Reviewer's Responses to Questions

**Comments to the Author**

1. Is the manuscript technically sound, and do the data support the conclusions?

Reviewer #1: Yes

Reviewer #2: Yes

2. Has the statistical analysis been performed appropriately and rigorously? 

Reviewer #1: No

Reviewer #2: Yes

3. Have the authors made all data underlying the findings in their manuscript fully available?

Reviewer #1: Yes

Reviewer #2: Yes

4. Is the manuscript presented in an intelligible fashion and written in standard English?

Reviewer #1: Yes

Reviewer #2: Yes

5. Review Comments to the Author

Reviewer #1: 1. Introduction

what is the significance of studying the quality of life breast cancer patients? since there are studies conducted before, what are you going to add to the scientific research? Have you added a new variable that was not studied before? these questions should be answered in the statement of the problem part

In addition, the statement of the problem needs to be updated, there are studies conducted with similar title in different pars of Ethiopia, so you need to incorporate those studies here.

2. sample size calculation

why have you used a "p" value from other country when there is a study conducted in Ethiopia? don't you think a study conducted with similar social background is preferable than other? besides, since the p value is far from 50%, you had to use "d (margin of error)" as 4 or 5, so that the sample size will be large

you should also state how many samples were taken from each hospital

3,data collection procedure

You have stated "However; a study in Ethiopia, Black Lion Specialised and Referral Hospital, dichotomized each scale as good or poor QOL, and We have followed the same classification for the current study" This study has classified the outcome variable arbitrarily with out a base reference. so it should not be used as reference. you shouldn't follow blindly just because it has been published elsewhere. so i do not agree with your outcome cutoff point, since it has no any base reference which can convince the reader why "75" is a cutoff for good or bad quality of life. besides why do you prefer classifying the outcome variable in two? why not analyse it with linear regression as it is, since transforming a variable results in a loss of information?

4. Analysis

you have used the other components of QLQ-C-30 other than global health component as an independent variable. Don't you think there is an interaction between these components as they are measuring or referring the same outcome? I suggest fitting different model for each functional scale components

5. Discussion

your discussion needs to be updated, since there are articles published in Ethiopia after 2019.

Reviewer #2: Please kindly find my comments below

Title: since there are studies conducted on quality of life of patients with breast cancer in Ethiopia, anything new to be added for the scientific world from this study? the authors the justification in your study, its new findings??

Abstract: In the result section of your abstract, it is good to put the 95%CI with magnitude of QOL.

Materials and Methods

Sample size and sampling technique

- What are the source and study populations, p/s clearly mention your source and study populations?

- Write the data collection period?

- make sure that the data collection tool is validated in Ethiopia or similar settings?

Result

- Good to avoid the first paragraph on the result section sub-title “Factors affecting the quality of life of breast cancer patient”:

- unrelated graph is annexed at the end of the document , p/s avoid it

6. PLOS authors have the option to publish the peer review history of their article (what does this mean?). If published, this will include your full peer review and any attached files.

Reviewer #1: No

Reviewer #2: **Yes: **Aragaw Tesfaw

---

## [Author Response · Author response to Decision Letter 0]

5 Dec 2022

Response to academic editor

Comment #1: Please ensure that your manuscript meets PLOS ONE's style requirements, including those for file naming. The PLOS ONE style templates can be found at

https://journals.plos.org/plosone/s/file?id=wjVg/PLOSOne_formatting_sample_main_body.pdf andhttps://journals.plos.org/plosone/s/file?id=ba62/PLOSOne_formatting_sample_title_authors_affiliations.pdf

Response#1: we thank you very much. We have double checked our revised paper. The current submission has been revised according to the journal style requirements.

Comment #2: Thank you for stating the following financial disclosure:

“Not applicable”

Response #2: Thank you very much. The authors received no specific funding for this work. Now, we have clearly indicated in the cover letter.

Comment #3: We note that you have indicated that data from this study are available upon request. PLOS only allows data to be available upon request if there are legal or ethical restrictions on sharing data publicly. For more information on unacceptable data access restrictions, please see http://journals.plos.org/plosone/s/data-availability#loc-unacceptabledata-access-restrictions.

Response #3: we uploaded the study’s minimal underlying dataset as Supporting Information files in the revised submission.

Comment #4: Please amend either the title on the online submission form (via Edit Submission) or the title in the manuscript so that they are identical.

Response #4: Thank you very much, this was corrected.

Comment #5: Your ethics statement should only appear in the Methods section of your manuscript. If your ethics statement is written in any section besides the Methods, please move it to the Methods section and delete it from any other section. Please ensure that your ethics statement is included in your manuscript, as the ethics statement entered into the online submission form will not be published alongside your manuscript.

Response #5: Thank you for your comment. In the current submission, the ethics statement is moved to the method section.

Comment #6: Please include a separate caption for each figure in your manuscript

Response #6: Thank you for suggesting editorial changes. We have revised the manuscript as per the comments.

Comment #7: Please include captions for your Supporting Information files at the end of your manuscript, and update any in-text citations to match accordingly. Please see our Supporting Information guidelines for more information: http://journals.plos.org/plosone/s/supporting-information.

Response #7: Thank you very much. This was corrected.

Comment #8: Please review your reference list to ensure that it is complete and correct. If you have cited papers that have been retracted, please include the rationale for doing so in the manuscript text, or remove these references and replace them with relevant current references. Any changes to the reference list should be mentioned in the rebuttal letter that accompanies your revised manuscript. If you need to cite a retracted article, indicate the article’s retracted status in the

References list and also include a citation and full reference for the retraction notice.

Response #8: Thank you very much for taking some spare time to review our paper. We thoroughly reviewed the reference list given in the manuscript, they are correct and complete.

Response to reviewer #2

1. Introduction

Comment #1: what is the significance of studying the quality of life breast cancer patients? Since there are studies conducted before, what are you going to add to the scientific research? Have you added a new variable that was not studied before? These questions should be answered in the statement of the problem part

In addition, the statement of the problem needs to be updated, there are studies conducted with similar title in different parts of Ethiopia, so you need to incorporate those studies here.

Response #1: Thank you for your comment. In fact, there are some recently added body of literature when the findings of this study on process for being published. Despite this fact, our study is a multicenter study which considered main hospitals with oncology centers in the Amhara region. Besides, it considered breast cancer patients at various treatment cycles and types, and various stages of the disease. These all gave us the chance to observe a broad picture of the QoL issues in the Amhara region of Ethiopia. While previous studies included only patients on chemotherapy treatment and almost all have used s generic health-related quality of life questionnaires originally developed by the World health Organization. Disease specific quality of life measurement is more specific and sensitive. The current study used a disease-specific QoL scale. In the assessment of patients’ QoL, there is evidence remarking that disease-specific QoL scales are preferred because they are sensitive and are capable of detecting small but clinically significant changes in health[1]. 

Comment #2: Sample size calculation

Why have you used a "p" value from other country when there is a study conducted in Ethiopia? Don’t you think a study conducted with similar social background is preferable than other? besides, since the p value is far from 50%, you had to use "d (margin of error)" as 4 or 5, so that the sample size will be large you should also state how many samples were taken from each hospital

Response #2: Thank you for your constructive comments. In fact, the principles of sample size calculation for cross sectional study (prevalence study); the sociodemographic characteristics should be same and updated “p” value shall consider. Despite this fact, in Ethiopia there were no previous literature and most of the developing countries quality of life score is similar. That is why we consider other countries. In fact, there are some recently added body of literature with a ”p” value in Ethiopia; 54.67%, 68.1% in Gondar and Black Lion hospital respectively[2, 3]. This is similar to other developing countries[4]. Finally using other country prevalence motioned as a limitation of this study.

In 2018/19, there were 217, 191, and 168 breast cancer patients on treatment or post-treatment follow-up in FHCSH, GRTH, and DRH, respectively. The sample was taken proportionally from three hospitals 98, 82, 76 in FHCSH, GRTH, and DRH, respectively.

One of the aims of applying appropriate sample size calculation formula is not to obtain the biggest sample size ever. The aim is to get an optimum or adequate sample size. Unnecessarily

large sample is not cost-effective. Investigators generally ends up with the ball-park figures of the study sizes usually based on their limitations such as financial resources, time or availability of subjects. It is appropriate to have a precision of 5% if the prevalence of the disease is in between 10% and 90%. This precision will give the width of 95% CI as 10% (e.g. 30% to 40%, or 60% to 70%). However, when the prevalence is below 10% or more than 90%, the precision of 5% seems to be inappropriate. For example, if the prevalence is 1% (in a rare disease) the precision of 5% is obviously crude and it may cause problems. Therefore, authors recommend margin of error (d) as a half of P if P is below 0.1 (10%) and if P is above 0.9 (90%), d can be {0.5(1-P)}. For example, if P is 0.04, investigators may use d=0.02, and if P is 0.98, may use d=0.01. Generally; if the prevalence within 10%-90%, it is safe to apply the ‘d=0.5’ suggestion[5].

Comment #3: Data collection procedure

You have stated "However; a study in Ethiopia, Black Lion Specialised and Referral Hospital, dichotomized each scale as good or poor QOL, and We have followed the same classification for the current study" This study has classified the outcome variable arbitrarily without a base reference. So it should not be used as reference. You shouldn't follow blindly just because it has been published elsewhere. So I do not agree with your outcome cutoff point, since it has no any base reference which can convince the reader why "75" is a cutoff for good or bad quality of life. besides why do you prefer classifying the outcome variable in two? Why not analyse it with linear regression as it is, since transforming a variable results in a loss of information?

Response #3: Thank you very much. As reviewer mentioned, the use of continuous variables is statistically preferable to the use of categorized variables in prognostic studies [6]. However, in order to be used easily in routine staging, QoL measures must be subdivided into more discrete categories; physicians usually based their decisions on a binary normal/abnormal assessment. Median, percentiles, or other arbitrary values have been selected as cutoffs for dichotomization into good or poor prognosis in the majority of studies[7]. The simplified interpretation of QoL can be achieved only through loss of information because values close to the cutoff point but in opposite directions are treated as equally different as the minimum and maximum values of the continuous variable. Furthermore, a cutoff point equal to the median value (which is not necessarily the optimal cutoff point) is equivalent to losing one-third of the data, resulting in loss of statistical power [6]. Still there are previous studies in Ethiopia recommending 75 as a cutoff points[8, 9].

Comment #4: Analysis

You have used the other components of QLQ-C-30 other than global health component as an independent variable. Don't you think there is an interaction between these components as they are measuring or referring the same outcome? I suggest fitting different model for each 

Response #4: Thank you very much. The EORTC QLQ-C30 is composed of 9 multi-item scales: 5 functioning scales (physical, role, cognitive, emotional and social), a global health status(QoL) scale, and 3 symptom scales (fatigue, pain and nausea/vomiting) and single items (dyspnea, insomnia, appetite loss, constipation, diarrhea and financial difficulties)[10]. 

The global health status (QoL) had two questions, with a modified 7-point linear analog scale ranging from 1 “very poor” to 7 “excellent”. All other items are scored on a 4-point categorical scale ranging from 1 “not at all” to 4 “very much” (Please see revised manuscript, page 9, line 159-161). 

QLQ-C-30 measuring independently the global health status and other functional component outcomes. Poor quality of life does not necessarily mean affected functional components and good functional component does not mean good quality of life. QoL is an individual’s overall satisfaction with life and general sense of personal well-being. QoL is patient perception of their position in life in the context of the culture and value systems in which they live and in relation to their goals, expectations, standards, and concerns.

Comment #5: Discussion

Your discussion needs to be updated, since there are articles published in Ethiopia after 2019.

Response #5: Thank you very much, this was updated.

Response to reviewer#1

Comment #1: Title: since there are studies conducted on quality of life of patients with breast cancer in Ethiopia, anything new to be added for the scientific world from this study? the authors the justification in your study, its new findings??

Response #1: thank you for the comments. Even though in Ethiopia breast cancer is among the leading causes of morbidity and mortality among women, little published studies have been conducted so far on the QoL in breast cancer women [11]. In Black Lion Hospital, a study was conducted to assess the QoL and associated factors among patients with breast cancer under chemotherapy. The Black Lion study included only patients under chemotherapy[11]. The inclusion of patients on various treatment types such as surgery, chemotherapy, hormonal therapy, and radiation therapy is important to have the big pictures of QoL concerns. Therefore, this study is probably the first study performed to measure the QoL and associated factors among breast cancer patients undergoing different forms of treatment in the Amhara region using the European Organization for Research and Treatment of Cancer Quality of Life Questionnaire Core 30(EORTC QLQ C30) and breast cancer supplementary measure (QLQ-BR23). 

Comment #2: Abstract: In the result section of your abstract, it is good to put the 95%CI with magnitude of QOL.

Response #2: Thank you very much; we have revised as per the comment.

Materials and Methods

Sample size and sampling technique

Comment #3: What are the source and study populations, p/s clearly mention your source and study populations?

Response #3: Thank you very much. Source population: All breast cancer patients being evaluated and treated in oncology units of three hospitals were considered as a source population. Study population: Those breast cancer patients who visit the hospitals and being evaluated or treated at the oncology units during data collection time (from March 25 to July 7, 2019) and who met the inclusion criteria were considered as a study population (Please see the revised document page 6, line 102 to 105).

Comment #4: Write the data collection period?

Response #4: The data collection was from March 25 to July 7, 2019 (Please see revised document, page 6, line 104).

Comment #5: make sure that the data collection tool is validated in Ethiopia or similar settings?

Response #5: To measure patients’ QoL, disease-specific QoL scales are preferred because they are sensitive and are capable of detecting small but clinically significant changes in health[1]. For this study, QoL scores were evaluated by using the Amharic version of EORTC version 3 of QLQ-C30 and its breast cancer supplementary measure (QLQ-BR23). It is a reliable and valid measure of QoL of cancer patients; the internal consistency was 0.81. The internal consistency of the subscales were greater than 0.73 except for cognitive function which was 0.29[12] (Please see the revised document page 8, line 139-150). 

The internal consistency of the EORTC QLQ was evaluated using the reliability coefficient (i.e. Cronbach's alpha value). The Cronbach's alpha value of EORTC QLQ-C30 and QLQ-BR23 was 0.80. The reliability coefficient of each subscale was also greater than 0.7 except for the cognitive function (0.63) and pain (0.65) subscales (Please see the revised document page 10, line 177-179).

Result

Comment #6: Good to avoid the first paragraph on the result section sub-title “Factors affecting the quality of life of breast cancer patient”:

Response #6: Thank you very much, this is corrected.

Comment #7: Unrelated graph is annexed at the end of the document, p/s avoid it

Response #7: thank you very much. This was editorial problem and now replaced with correct graph.

Reference

1. Davies N, G.E., Mackintosh A, Fitzpatrick, A Structured Review of Patient-Reported Outcome Measures for Women with Breast Cancer. 2009.

2. Tadesse Melaku Abegaz , A.A.A., and Begashaw Melaku Gebresillassie, Health Related Quality of Life of Cancer Patients in Ethiopia. . Hindawi Journal of Oncology, 2018: p. 8.

3. Selamawit Gebrehiwot Sibhat, T.G.F., Beate Sander, and Gebremedhin Beedemariam Gebretekle, Health-related quality of life and its predictors among patients with breast cancer at Tikur Anbessa Specialized Hospital, Addis Ababa, Ethiopia. Health and Quality of Life Outcomes 2019. 17: p. 165.

4. Kanayamkandi J, S.S., Quality of life among breast cancer patients: a cross sect

---

## [Decision Letter · Decision Letter 1]

9 Aug 2023

PONE-D-21-10111R1Quality of life of breast cancer patients in Amhara region, Ethiopia: A cross-sectional studyPLOS ONE

Dear Dr. Alem,

Thank you for submitting your manuscript to PLOS ONE. After careful consideration, we feel that it has merit but does not fully meet PLOS ONE’s publication criteria as it currently stands. Therefore, we invite you to submit a revised version of the manuscript that addresses the points raised during the review process.

ACADEMIC EDITOR:I suggest language revision of the manuscript.

We look forward to receiving your revised manuscript.

Kind regards,

Omnia Samir El Seifi, Professor

Academic Editor

PLOS ONE

Journal Requirements:

Reviewers' comments:

Reviewer's Responses to Questions

**Comments to the Author**

1. If the authors have adequately addressed your comments raised in a previous round of review and you feel that this manuscript is now acceptable for publication, you may indicate that here to bypass the “Comments to the Author” section, enter your conflict of interest statement in the “Confidential to Editor” section, and submit your "Accept" recommendation.

Reviewer #2: All comments have been addressed

2. Is the manuscript technically sound, and do the data support the conclusions?

Reviewer #2: Yes

3. Has the statistical analysis been performed appropriately and rigorously? 

Reviewer #2: Yes

4. Have the authors made all data underlying the findings in their manuscript fully available?

Reviewer #2: (No Response)

5. Is the manuscript presented in an intelligible fashion and written in standard English?

Reviewer #2: Yes

6. Review Comments to the Author

Reviewer #2: (No Response)

7. PLOS authors have the option to publish the peer review history of their article (what does this mean?). If published, this will include your full peer review and any attached files.

Reviewer #2: **Yes: **Aragaw Tesfaw

---

## [Author Response · Author response to Decision Letter 1]

30 Dec 2023

we thank you very much. We tried to edit the entire document of the manuscript. We edited the spelling, space between words, and the whole grammar as much as possible with online grammar and language checkers (Quill bot online checker).

Thank you very much. We uploaded the study’s minimal underlying dataset as Supporting Information files.

We have revised the whole manuscript for dual publication and research ethics.

---

## [Decision Letter · Decision Letter 2]

30 Jan 2024

PONE-D-21-10111R2Quality of life of breast cancer patients in Amhara region, Ethiopia: A cross-sectional studyPLOS ONE

Dear Dr. Alem,

Thank you for submitting your manuscript to PLOS ONE. After careful consideration, we feel that it has merit but does not fully meet PLOS ONE’s publication criteria as it currently stands. Therefore, we invite you to submit a revised version of the manuscript that addresses the points raised during the review process.

We look forward to receiving your revised manuscript.

Kind regards,

Omnia Samir El Seifi, M.D.

Academic Editor

PLOS ONE

Journal Requirements:

Reviewers' comments:

Reviewer's Responses to Questions

**Comments to the Author**

1. If the authors have adequately addressed your comments raised in a previous round of review and you feel that this manuscript is now acceptable for publication, you may indicate that here to bypass the “Comments to the Author” section, enter your conflict of interest statement in the “Confidential to Editor” section, and submit your "Accept" recommendation.

Reviewer #3: (No Response)

2. Is the manuscript technically sound, and do the data support the conclusions?

Reviewer #3: Yes

3. Has the statistical analysis been performed appropriately and rigorously? 

Reviewer #3: Yes

4. Have the authors made all data underlying the findings in their manuscript fully available?

Reviewer #3: (No Response)

5. Is the manuscript presented in an intelligible fashion and written in standard English?

Reviewer #3: (No Response)

6. Review Comments to the Author

Reviewer #3: This manuscript reports results from a cross-sectional study investigating quality of life of breast cancer patients in Amhara region of Ethiopia. I have below comments.

Line 112, the sample size calculation, how does 248 come from? Based on the given information, n should be 246. Considering the 10% non-response rate, the final sample size 273 should be obtained by 246/0.9=273.

Line 207, there are only 256 patients with a response rate of 93.4%. Please explain why the recruitment of patients did not meet the designed sample size of 273? From 256 with a response rate of 93.4%, which means only 239 patients have response, it is less than the target sample size n=246. Please discuss if it will it affect the results.

In Table 4, what does COR or AOR stand for? Please add notes corresponding to them. It is also not clear about the QoL. Is it functional QoL, symptom QoL, or global QoL?

7. PLOS authors have the option to publish the peer review history of their article (what does this mean?). If published, this will include your full peer review and any attached files.

Reviewer #3: No

---

## [Author Response · Author response to Decision Letter 2]

18 Feb 2024

The study achieved beyond the minimum sample size required in planning stage of the study. We interviewed 256 patients while the minimum sample size required was 246. The study achieved 93.8% response rate, which could be obtained by dividing the actual response divided by the plan (256/273 = 93.8%). The actual non-response rate is below the contingency plan of 10%. Thus, the number of patients included in the study are adequate and have not any negative effect on the results.

There is no single definition of QoL but scholars define it in different ways; it is “the subjective evaluation of the good and satisfactory character of life as a whole”[1].

Symptom QoL relates to the individual's experience of symptoms or discomfort associated with breast cancer.

Global health (QoL) reflects an overall assessment of an individual's satisfaction and happiness with their life as a whole.

Functional QoL: refers to the individual's ability to perform daily activities and tasks related to their physical, emotional, and social well-being, despite the challenges posed by the disease and its treatment.

The EORTC QLQ-C30 is a tool used to address quality of life issues to all cancer type patients and it is composed of nine multi-item scales and six single items. The multi-item scales include five functioning scales (physical, role, cognitive, emotional, and social), a global health status (QoL) scale, and three symptom scales (fatigue, pain, and nausea/vomiting). The six single items included dyspnea, insomnia, appetite loss, constipation, diarrhea, and financial difficulties [3]. The EORTC QLQ-BR23 is unique to breast cancer patients and it is composed of four functional scales (future perspective, body image, sexual function, and sexual enjoyment) and four symptom scales (systemic therapy side effect, arm symptom, breast symptom, financial difficulties, and upset by hair loss).

---

## [Decision Letter · Decision Letter 3]

26 Feb 2024

PONE-D-21-10111R3Quality of life of breast cancer patients in Amhara region, Ethiopia: A cross-sectional studyPLOS ONE

Dear Dr. Alem,

Thank you for submitting your manuscript to PLOS ONE. After careful consideration, we feel that it has merit but does not fully meet PLOS ONE’s publication criteria as it currently stands. Therefore, we invite you to submit a revised version of the manuscript that addresses the points raised during the review process. 

We look forward to receiving your revised manuscript.

Kind regards,

Omnia Samir El Seifi, M.D.

Academic Editor

PLOS ONE

Journal Requirements:

Reviewers' comments:

Reviewer's Responses to Questions

Comments to the Author

1. If the authors have adequately addressed your comments raised in a previous round of review and you feel that this manuscript is now acceptable for publication, you may indicate that here to bypass the “Comments to the Author” section, enter your conflict of interest statement in the “Confidential to Editor” section, and submit your "Accept" recommendation.

Reviewer #3: (No Response)

2. Is the manuscript technically sound, and do the data support the conclusions?

Reviewer #3: (No Response)

3. Has the statistical analysis been performed appropriately and rigorously? 

Reviewer #3: (No Response)

4. Have the authors made all data underlying the findings in their manuscript fully available?

Reviewer #3: (No Response)

5. Is the manuscript presented in an intelligible fashion and written in standard English?

Reviewer #3: (No Response)

6. Review Comments to the Author

Reviewer #3: For the QoL, it sounds that Table 4 used QLQ-BR23 to determine subject’s QoL outcome as good or poor QoL. Since QLQ-BR23 includes both functional and symptom items, please provide details about how to assign the dichotomous outcome to each subject. E.g. for one subject, if her average functional scale (>=75) is good, but average symptom scale (>=25) is poor, will this subject’s QoL be considered as poor?

7. PLOS authors have the option to publish the peer review history of their article (what does this mean?). If published, this will include your full peer review and any attached files.

Do you want your identity to be public for this peer review? For information about this choice, including consent withdrawal, please see our Privacy Policy.

Reviewer #3: No

---

## [Author Response · Author response to Decision Letter 3]

7 May 2024

We appreciate for the comment. The EORTC QLQ-BR23 is unique to breast cancer patients and it is composed of four functional scales (future perspective, body image, sexual function, and sexual enjoyment) and four symptom scales (systemic therapy side effect, arm symptom, breast symptom, and upset by hair loss). The functional scale assesses various aspects related to the patient’s well-being, and the symptom scale evaluates symptoms experienced by the patient. The breast cancer tools incorporates five multi-item scales to assess systemic therapy side effects, arm symptoms, breast symptoms, body image and sexual functioning. In addition, single items assess sexual enjoyment, hair loss and future perspective [1]. 

The quality of life (QoL) of breast cancer patients had two global health status questions, with a modified 7-point linear analog scale ranging from 1 “very poor” to 7 “excellent”. For one subject, if her average functional scale (>=75) is good, and average symptom scale (>=25) is poor, will this subject’s QoL might be good or poor. The patients QoL determined by the result of global health status scores, without considering the functional and symptoms scale results (See below Table 1).

---

## [Decision Letter · Decision Letter 4]

28 May 2024

Quality of life of breast cancer patients in Amhara region, Ethiopia: A cross-sectional study

PONE-D-21-10111R4

Dear Dr. Alem,

We’re pleased to inform you that your manuscript has been judged scientifically suitable for publication and will be formally accepted for publication once it meets all outstanding technical requirements.

Kind regards,

Omnia Samir El Seifi, M.D.

Academic Editor

PLOS ONE

Additional Editor Comments (optional):

Reviewers' comments:

Reviewer's Responses to Questions

**Comments to the Author**

1. If the authors have adequately addressed your comments raised in a previous round of review and you feel that this manuscript is now acceptable for publication, you may indicate that here to bypass the “Comments to the Author” section, enter your conflict of interest statement in the “Confidential to Editor” section, and submit your "Accept" recommendation.

Reviewer #3: All comments have been addressed

2. Is the manuscript technically sound, and do the data support the conclusions?

Reviewer #3: (No Response)

3. Has the statistical analysis been performed appropriately and rigorously? 

Reviewer #3: (No Response)

4. Have the authors made all data underlying the findings in their manuscript fully available?

Reviewer #3: (No Response)

5. Is the manuscript presented in an intelligible fashion and written in standard English?

Reviewer #3: (No Response)

6. Review Comments to the Author

Reviewer #3: (No Response)

7. PLOS authors have the option to publish the peer review history of their article (what does this mean?). If published, this will include your full peer review and any attached files.

Reviewer #3: No

---

## [Editor Report · Acceptance letter]

20 Jun 2024

PONE-D-21-10111R4 

PLOS ONE

Dear Dr. Alem, 

I'm pleased to inform you that your manuscript has been deemed suitable for publication in PLOS ONE. Congratulations! Your manuscript is now being handed over to our production team.

Kind regards, 

on behalf of

Professor Omnia Samir El Seifi 

Academic Editor

PLOS ONE